# Discovering Temporally-Aware Reinforcement Learning Algorithms

**Matthew T. Jackson**[*]
University of Oxford

**Chris Lu**[*]
University of Oxford

**Louis Kirsch**
The Swiss AI Lab IDSIA

**Robert T. Lange**
Technical University Berlin

**Shimon Whiteson**
University of Oxford

**Jakob N. Foerster**
University of Oxford

## Abstract

Recent advancements in meta-learning have enabled the automatic discovery of novel reinforcement learning algorithms parameterized by surrogate objective functions. To improve upon manually designed algorithms, the parameterization of this learned objective function must be expressive enough to represent novel principles of learning (instead of merely recovering already established ones) while still generalizing to a wide range of settings outside of its meta-training distribution. However, existing methods focus on discovering objective functions that, like many widely used objective functions in reinforcement learning, do not take into account the total number of steps allowed for training, or "training horizon". In contrast, humans use a plethora of different learning objectives across the course of acquiring a new ability. For instance, students may alter their studying techniques based on the proximity to exam deadlines and their self-assessed capabilities. This paper contends that ignoring the optimization time horizon significantly restricts the expressive potential of discovered learning algorithms. We propose a simple augmentation to two existing objective discovery approaches that allows the discovered algorithm to dynamically update its objective function throughout the agent's training procedure, resulting in expressive schedules and increased generalization across different training horizons. In the process, we find that commonly used meta-gradient approaches fail to discover such adaptive objective functions while evolution strategies discover highly dynamic learning rules. We demonstrate the effectiveness of our approach on a wide range of tasks and analyze the resulting learned algorithms, which we find effectively balance exploration and exploitation by modifying the structure of their learning rules throughout the agent's lifetime.

## 1 Introduction

Advancements in reinforcement learning (RL) have historically come from handcrafted algorithms derived from human experimentation or theoretical insights (Mnih et al., 2015; Schulman et al., 2015). By contrast, recent approaches attempt to discover novel RL algorithms automatically (e.g. Kirsch et al., 2020; Oh et al., 2020; Lu et al., 2022a). One well-studied feature of handcrafted RL algorithms is the adaptation of the update to the total and remaining number of training steps (Auer, 2002; Farsang & Szegletes, 2021). This adaptation to perceived learning horizon is even observed in humans, for instance, students using spaced repetition systems with decreasing intervals to maximize learning progress (Smith & Scarf, 2017).

In this work, we take a meta-learning approach to discover novel RL objective functions that are aware of and adapt to the amount of learning time remaining. We modify Learned Policy Gradient (Oh et al., 2020, LPG) and Learned Policy Optimization (Lu et al., 2022a, LPO), two recent meta-learned objective methods, to directly condition on temporal information about the agent's lifetime. We find the meta-optimization method is a critical component in successfully taking advantage of this information, which we refer to as "lifetime conditioning". Specifically, meta-gradient

---

[*]Equal contribution. Correspondence to jackson@robots.ox.ac.uk; christopher.lu@exeter.ox.ac.uk.

approaches using a myopic proxy objective (as in LPG) fail to learn a dynamic update. Instead, we investigate the use of Evolution Strategies (Salimans et al., 2017, ES), which computes fitness from the final return after the *entire lifetime of the RL agent*. These additions enable both of the methods to discover objective functions that dynamically adapt across the course of learning.

We evaluate the learned objective functions on both in-distribution and out-of-distribution environments, ranging from continuous control tasks to discrete Atari-like settings, over a range of training horizons. They significantly improve upon the performance of their non-temporally-aware counterparts, improving generalization to previously unseen training horizons and environments. Our results highlight the synergistic effects of combining lifetime conditioning with ES when meta-learning temporally-aware objective functions.

Our contributions are summarized as follows:

1. We propose *temporally-aware* variants of LPG and LPO, named TA-LPG and TA-LPO. We augment their input spaces to contain temporal information, allowing for direct conditioning on the total and remaining training horizon of the agent (Section 4.1).

2. We compare meta-gradient and evolutionary meta-optimization approaches for discovering non-myopic RL objective functions, finding that meta-gradient approaches fail to learn temporally-aware updates (Section 5.3). Based on this, we propose the use of ES for TA-LPG, with a simple adaptation for multi-task meta-learning (Section 4.2).

3. After meta-training, we test the discovered objective functions over a range of evaluation environments. Our temporally-aware objective functions outperform all baselines we consider, generalizing to environments and training horizons unseen during meta-training (Section 5.2).

4. We analyze the dynamic schedules extracted by the discovered objective functions. We find that they implement dynamic policy importance ratio clipping, in addition, to update and entropy annealing schedules that effectively adapt to the training horizon, thereby balancing exploration and exploitation (Section 5.3).

## 2 RELATED WORK

**Meta-learning fundamentals** Meta-learning (or "learning to learn") aims to discover new principles of learning via end-to-end optimization (Schmidhuber, 1987; Thrun & Pratt, 1998). These approaches can be grouped into meta-gradient (Finn et al., 2017; Xu et al., 2018) and memory-based (in-context learning) approaches (Hochreiter et al., 2001; Wang et al., 2016; Duan et al., 2016). Instead of training agents on a single task instance, meta-learning discovers parameterized learning algorithm components that generalize across a task distribution. Two key related considerations arise: First, how flexible should the meta-optimized medium be? Suitable inductive biases constrain the meta-search space, which can facilitate stable and more data-efficient meta-optimization. However, this comes at the risk of prohibiting truly novel algorithm discovery. Second, how can we avoid overfitting to the meta-training task distribution (Kirsch et al., 2020; Jackson et al., 2023) and rollout horizon (Lange & Sprekeler, 2022) to find truly general-purpose components instead of heuristics? In this work, we argue that both of these core problems can be tackled by making the learned algorithm aware of the learning horizon of the agent.

**Meta-learning objective functions** One specific approach to meta-learning is concerned with the meta-optimization of parameterized objective functions to represent learning algorithms (Houthooft et al., 2018; Bechtle et al., 2021). More recently, several advances were achieved in order to facilitate generalization. These include, e.g., MetaGenRL (Kirsch et al., 2020), LPG (Oh et al., 2020), LPO (Lu et al., 2022a) and GROOVE (Jackson et al., 2023). Finally, several attempts have been made to obtain interpretable RL algorithm (components) using, for example, symbolic methods (Alet et al., 2020; Co-Reyes et al., 2021). Here, we leverage LPG and LPO to demonstrate that further performance improvements can be achieved by giving the algorithm access to its current position within its total lifetime.

**Meta-optimization with Evolutionary / Zeroth-order methods** Previously, it has been observed that meta-gradient approaches allow only for short inner loop algorithm unrolls due to memory

constraints and chaotic dynamics (Metz et al., 2021; Wu et al., 2018). This in turn can lead to short-sightedness (myopia) of the discovered element. An alternative approach is to perform black-box optimization using evolutionary optimization. This circumvents the computation of higher-order gradients and accommodates full algorithm rollouts. This has been successfully applied in the context of learning gradient-based (Metz et al., 2022) and gradient-free optimizers (Lange et al., 2023a;b) as well as in-context learning (Kirsch & Schmidhuber, 2021), adversarial learning (Lu et al., 2022b; 2023b) and neural auto-curricula (Feng et al., 2021).

## 3 BACKGROUND

### 3.1 META REINFORCEMENT LEARNING

Discovering new reinforcement learning algorithms is a form of meta-reinforcement learning (Schmidhuber, 1987; Parker-Holder et al., 2022). However, most meta-reinforcement learning approaches aim to adapt to a small range of tasks (Duan et al., 2016; Wang et al., 2016; Finn et al., 2017; Houthooft et al., 2018; Bechtle et al., 2021) such as varying rewards signals or environment parameterizations. Here, we follow a line of work that aims to meta-learn reinforcement learning algorithms that can learn across a wide range of reinforcement learning environments (Kirsch et al., 2020; Oh et al., 2020; Lu et al., 2022a) such as generalizing to robotic control environments of different actuators and environment dynamics or from simple grid worlds to robotic control problems.

More formally, in meta-RL we search for learning algorithms that solve reinforcement learning (RL) problems, in the simplest case defined as Markov Decision Processes (MDPs). An MDP is defined by the tuple $\langle \mathcal{S}, \mathcal{A}, R, P, \gamma, d \rangle$. At each timestep $t$, an agent takes an action sampled from its policy $a_t \sim \pi(\cdot|s_t)$ (where $a_t \in \mathcal{A}$ and $s_t \in \mathcal{S}$). The environment then returns a reward $R(s_t, a_t)$ and samples the next state $s_{t+1}$ according to the transition function $s_{t+1} \sim P(\cdot|s_t, a_t)$. The objective is to find a policy $\pi$ (parameterized by $\theta$) that maximizes the expected return:

$$J(\pi_\theta) \triangleq \mathbb{E}[R^\gamma|\pi_\theta] = \mathbb{E}_{s_0 \sim d, a_{0:\infty} \sim \pi_\theta, s_{1:\infty} \sim P}\Big[ \sum_{t=0}^{\infty} \gamma^t R(s_t, a_t) \Big]. \tag{1}$$

In our approach, meta-optimization operates in a space of RL algorithms, represented by meta-parameters $\phi$, which are used to update the policy parameters $\theta$. For example, if $\phi$ parameterizes a loss function $L_\phi$ that we optimize with gradient descent (using learning rate $\eta$), then $\theta_{k+1} = \theta_k - \eta \nabla_\theta L_\phi(\theta_k)$. Let $\pi_{\theta_k}$ be the policy after $k$ updates to an initial policy $\pi_{\theta_0}$ with $\phi$. We maximize the expected return after $K$ updates, or at the end of agent training, which we define as:

$$F(\phi) = \mathbb{E}_{\theta_0}[J(\pi_{\theta_K})]. \tag{2}$$

In our work, we focus on instances of meta-RL that parameterize surrogate loss functions with $\phi$ and apply gradient-based updates to $\pi_\theta$ We focus on two recent instances thereof: Learned Policy Gradients (Oh et al., 2020, LPG) and Learned Policy Optimization (Lu et al., 2022a, LPO).

### 3.2 LEARNED POLICY GRADIENT (LPG)

Learned Policy Gradient (LPG) (Oh et al., 2020) is a meta-RL approach that generalizes actor-critic RL (Barto et al., 1983). Rather than training a critic to output a scalar value estimate, it trains a critic that outputs a *bootstrap vector*. The interpretation of this vector is meta-learned by LPG, allowing it to encode information beyond value estimation that helps it optimize the inner policy, e.g., state visitation.

LPG is parameterized by an LSTM (Hochreiter & Schmidhuber, 1997), which iterates over a reversed sequence of agent transitions. Specifically, to compute an update to the joint actor-critic parameters $\theta$ at timestep $t$, LPG outputs targets $\hat{y}_t, \hat{\pi}_t = U_\phi(x_t|x_{t+1}, \ldots, x_T)$. Here, the input vector $x_t$ is constructed from the transition $\tau_t = (s_t, a_t, r_t, d_t, s_{t+1})$ at timestep $t$ by

$$x_t = [r_t, d_t, \gamma, \pi_\theta(a_t|s_t), y_\theta(s_t), y_\theta(s_{t+1})], \tag{3}$$

containing reward $r_t$, episode-termination flag $d_t$, discount factor $\gamma$, probability of the chosen action $\pi_\theta(a_t|s_t)$, and bootstrap vectors for the current and next states $y_\theta(s_t)$ and $y_\theta(s_{t+1})$. The targets $\hat{y}$ and $\hat{\pi}$ update the critic and policy respectively, giving the update rule

$$\Delta\theta \propto \mathbb{E}_\tau \left[ \nabla_\theta \log \pi_\theta(a|s)\hat{\pi} - \alpha_y \nabla_\theta D_{\text{KL}}(y_\theta||\hat{y}) \right]. \tag{4}$$

LPG is optimized over a series of update steps using *meta-gradients*, to maximize the return of the policy at the end of training. Given a distribution of environments $\mathcal{E}$ and initial agent parameters $\theta_0$, these meta-gradients are computed by

$$\Delta\phi \propto \mathbb{E}_{\mathcal{E}}\mathbb{E}_{\theta_0}\left[\nabla_\phi \log \pi_{\theta_K}(a|s)J(\pi_{\theta_K})\right], \tag{5}$$

where $\pi_{\theta_K}$ is the policy produced from an initialization $\theta_0$ after updating with $U_\phi$ for $K$ steps. This computation requires backpropagation through the agent's entire learning process, making its computation limited by memory constraints. Therefore, a truncated backpropagation window is used in practice, which optimizes the agent over $k \ll K$ update steps.

## 3.3 LEARNED POLICY OPTIMISATION (LPO)

Learned Policy Optimisation (Lu et al., 2022a, LPO) is a meta-RL method that inherits theoretical convergence guarantees from the Mirror Learning framework (Kuba et al., 2022). In Mirror Learning, agents maximize

$$\mathbb{E}_{a\sim\pi_{\text{new}}}[Q_{\pi_{\text{old}}}(s,a)] - \mathfrak{D}_{\pi_{\text{old}}}(\pi_{\text{new}}|s) \tag{6}$$

where $Q_\pi(s,a) \triangleq \mathbb{E}[\mathrm{R}^\gamma|\pi, \mathrm{s}_0 = s, \mathrm{a}_0 = a]$ (the state-action value function of $\pi$) and $\mathfrak{D}$ is the *drift function*, which maps from two policies in a given state to a scalar. If $\mathfrak{D}$ satisfies specific conditions from Kuba et al. (2022),

1. It is non-negative everywhere and zero at identity $\mathfrak{D}_{\pi_k}(\pi|s) \geq \mathfrak{D}_{\pi_k}(\pi_k|s) = 0$,

2. Its gradient with respect to $\pi$ is zero at $\pi = \pi_k$,

then the Mirror Learning algorithm provably improves monotonically and converges to the optimal return in the limit. Many existing RL algorithms are instances of Mirror Learning. Most notably, PPO (Schulman et al., 2017) is an instance of Mirror Learning since its clipped objective can be reformulated as a drift function.

LPO parameterizes $\mathfrak{D}$ with a neural network and structures its inputs and network architecture to guarantee the drift function conditions hold. The network does not use bias parameters and takes the following vector as input:

$$\boldsymbol{x}_{p,A} = [(1-p),\ (1-p)^2,\ (1-p)A,\ (1-p)^2A,\ \log(p),\ \log(p)^2,\ \log(p)A,\ \log(p)^2A], \tag{7}$$

where $p$ is the probability ratio between the current policy and the original policy for a given state-action pair $(s,a)$ from the environment $p \triangleq \frac{\pi_{\text{new}}(a|s)}{\pi_{\text{old}}(a|s)}$ and $A$ is the estimated advantage of the state-action pair $A_\pi(s,a) \triangleq Q_\pi(s,a) - \mathbb{E}_{\hat{a}\sim\pi(\cdot|s)}[Q_\pi(s,\hat{a})]$. By removing the bias parameters and multiplying all inputs by $(1-p)$ or $\log(p)$, this input parameterization guarantees that the drift function conditions hold.

LPO uses evolution strategies (Salimans et al., 2017, ES) to optimize $\mathfrak{D}$. ES estimates the gradient of a smoothed black-box function from samples. Formally,

$$\nabla_{\mathbf{x}}\mathbb{E}_{\epsilon\sim N(\mathbf{0},I_d)}[F(\mathbf{x} + \sigma\epsilon)] = \mathbb{E}_{\epsilon\sim N(\mathbf{0},I_d)}\left[\frac{\epsilon}{\sigma}F(\mathbf{x} + \sigma\epsilon)\right]. \tag{8}$$

where $F$ is the objective defined in Equation 2.

## 4 METHOD

### 4.1 CONDITIONING ON AGENT LIFETIME

We define the current lifetime of the agent as $n/N$, where $n$ is the number of environment interactions the agent has had so far and $N$ is the total number of environment interactions it will have, i.e., the training horizon. By conditioning on this information, we propose two extensions to existing learned-loss meta-RL methods: *Temporally-Adaptive* Learned Policy Gradient (TA-LPG) and *Temporally-Adaptive* Learned Policy Optimization (TA-LPO).

**Temporally-Adaptive LPG**   We incorporate lifetime conditioning into LPG by appending the agent's lifetime and the training horizon to the optimizer input. Specifically, we take the logarithm of the training horizon $\log(N)$ due to this input being unbounded. This results in an input of

$$x_t = [r, d, \gamma, \pi_\theta(a|s), y_\theta(s), y_\theta(s), \frac{n}{N}, \log(N)]. \tag{9}$$

**Temporally-Adaptive LPO**   To condition LPO on an agent's lifetime, we provide the lifetime as input to $\mathfrak{D}$, the drift function network. We follow LPO's implementation and structure the input to guarantee that the theoretical drift function conditions from Equation 6 hold. In particular, we input

$$\boldsymbol{x}_{p,A,t} = [\boldsymbol{x}_{p,A}, \frac{n}{N}\boldsymbol{x}_{p,A}] \tag{10}$$

where $\boldsymbol{x}_{p,A}$ is defined in Equation 7. Since TA-LPO is only meta-trained on a single environment and horizon, we do not include the $\log(N)$ term since it does not vary across updates.

## 4.2   META-OPTIMIZATION FOR LIFETIME ADAPTATION

**Gradient-free vs. gradient-based meta-optimization**   A range of meta-RL methods, including LPG, compute the gradient of the meta-parameters in order to update the learned loss function during meta-training (Kirsch et al., 2020; Bechtle et al., 2021). When meta-optimizing over multiple update steps, these meta-gradient approaches require backpropagation through time (Werbos, 1990). This raises a range of stability issues, particularly early in meta-training when the update rule is unstable and the inner policy gradient may display chaotic dynamics (Metz et al., 2021). On top of this, the memory requirement severely limits the number of steps that can be backpropagated through, requiring the backpropagation window to be truncated to a reduced number of steps, typically far below the training horizon. This leads to a *myopic* proxy objective for meta-gradient methods, which may prevent them from effectively conditioning on an agent's lifetime. We investigate this in Section 5.3.

ES is an attractive alternative to backpropagation-based meta-gradients for meta-optimization, as it can optimize through an unlimited number of inner loop update steps with no additional memory requirement. This makes it possible to optimize over the entire lifetime of an agent, such that the objective becomes the agent's return at the end of training, rather than after a truncated series of updates. ES has previously been used for meta-learning objective functions in the single-task setting (Section 3.3). To avoid instabilities in the multi-task setting, we propose a simple adaptation to ES.

**Adapting ES to multi-task meta-learning**   Given a large or continuous distribution of tasks $\mathcal{E}$, it is infeasible to evaluate the expected fitness of each parameter candidate $\mathbb{E}_{e\sim\mathcal{E}}[F_e(\mathbf{x} + \sigma\epsilon)]$ across the entire task distribution, where $F_e$ is the fitness function on task $e$. In contrast, evaluating all candidates on an individual task $e \sim \mathcal{E}$ each update step leads to significant bias in the meta-update, since it is derived from a single task. To balance between these, we could evaluate each candidate $\epsilon_i$ on a distinct task $e_i \sim \mathcal{E}$. This balances these factors but can increase update variance, particularly when return is not normalized between tasks.

To solve this, we propose *antithetic task sampling*, which builds upon antithetic sampling (Geweke, 1988). In antithetic sampling, pairs of candidates $\mathbf{x} + \sigma\epsilon$ and $\mathbf{x} - \sigma\epsilon$ are evaluated for each sampled noise vector $\epsilon$, reducing update variance in practice. In the multi-task setting, this has the form

$$\mathbb{E}_{\epsilon\sim N(\mathbf{0},I_d)} \left[ \frac{\epsilon}{2\sigma} (\mathbb{E}_{e\sim\mathcal{E}}[F_e(\mathbf{x} + \sigma\epsilon)] - \mathbb{E}_{e\sim\mathcal{E}}[F_e(\mathbf{x} - \sigma\epsilon)]) \right]. \tag{11}$$

We adapt this method by evaluating each antithetic candidate pair on the *same randomly sampled task*, giving the estimator

$$\mathbb{E}_{\epsilon\sim N(\mathbf{0},I_d)} \left[ \mathbb{E}_{e\sim\mathcal{E}} \left[ \frac{\epsilon}{2\sigma} (F_e(\mathbf{x} + \sigma\epsilon) - F_e(\mathbf{x} - \sigma\epsilon)) \right] \right]. \tag{12}$$

In practice, we first apply a rank transformation over the pair's fitness, which is equivalent to selecting the higher-performing member. In doing so, we normalize across tasks by returning a uniform fitness from each task. We found this stabilized training and led to higher performance in preliminary experiments, so we adopted this approach for all multi-task ES optimization.

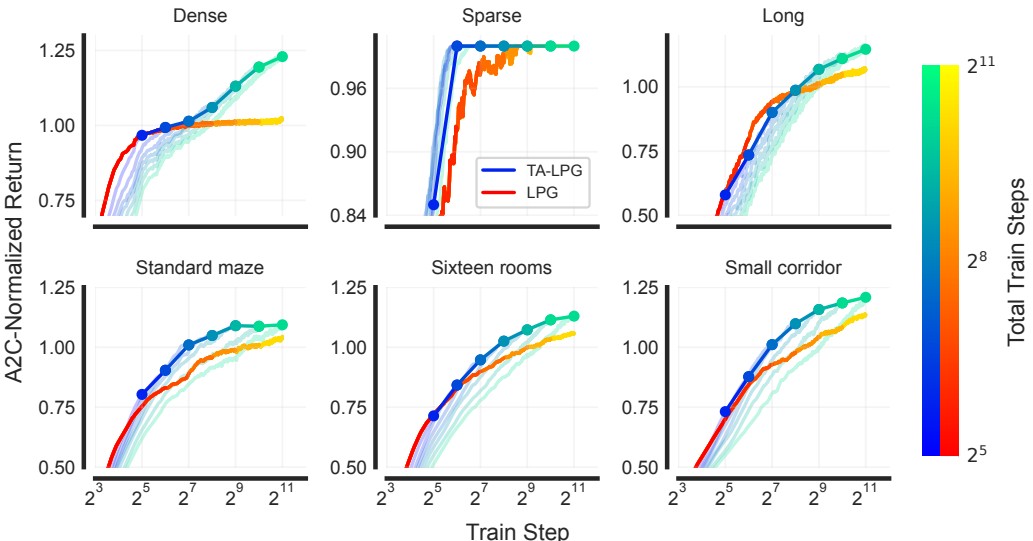

Figure 1: **TA-LPG adapts to variable training horizons.** Training curves for LPG and TA-LPG on held-out Grid-World environments from Oh et al. (2020) (top) and Minigrid (bottom), for a range of total train steps. Since TA-LPG adapts to the training horizon, we plot individual training curves for each horizon (faded lines) and their final return (bold points), with the color gradient reflecting the horizon for each model. We observe the final return of TA-LPG at each horizon is consistently greater than the LPG return at the same point. Return is normalized against an A2C agent trained to convergence and averaged over 5 meta-train and 128 meta-test seeds.

## 5 EXPERIMENTS

To investigate the effectiveness of lifetime conditioning, we evaluate TA-LPG and TA-LPO against their non-temporally-aware counterparts. Section 5.1 describes our experimental setup in more detail. In Section 5.2, we demonstrate improved performance from TA-LPG against LPG over a range of training horizons, before showing improved performance from TA-LPO against LPO on out-of-distribution environments and analyze TA-LPG and TA-LPO in Section 5.3.

### 5.1 EXPERIMENTAL SETUP

**Model architecture and training**   We use the reference model hyperparameters for both LPG and LPO (Appendix A). As discussed in Section 4.2, we use ES with antithetic task sampling for the meta-optimization of LPG, until Section 5.3, where we ablate ES against the original gradient-based meta-optimization method. As in the reference work, ES is used for LPO throughout. We implement the entire training process in JAX (Bradbury et al., 2018), using evosax (Lange, 2023) for evolution.

**Training environments**   In our experiments, we follow the meta-training environments originally used in Oh et al. (2020) and Lu et al. (2022a) for LPG and LPO respectively. LPG is meta-trained in a multi-task setting over a continuous distribution of Grid-World environments, with variable training horizons per task. In contrast, LPO is meta-trained on a single environment (MinAtar Space Invaders (Young & Tian, 2019; Lange, 2022), with a fixed training horizon. For this reason, we train and evaluate LPG for variable horizon adaptation, whilst we evaluate the use of relative time-step information with LPO.

**Testing environments**   At meta-test time, we evaluate LPO on the MinAtar and Brax (Freeman et al., 2021) evaluation suites, using the PPO hyperparameters from PureJaxRL (Lu et al., 2022a). Notably, while the hyperparameters are shared within each suite, they differ significantly between the two. We evaluate LPG on held-out Grid-World configurations from the reference LPG publication and Minigrid (Chevalier-Boisvert et al., 2023) across a range of training horizons.

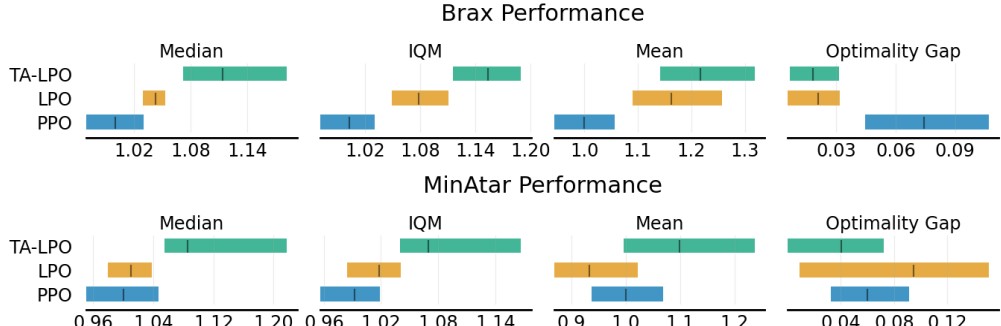

Figure 2: **TA-LPO leverages lifetime information and generalizes to a wide range of environments.** Results of TA-LPO, LPO and PPO on the Brax and MinAtar suites across three seeds. TA-LPO was only meta-trained on SpaceInvaders-MinAtar. We provide complete training curves in Appendix B.

## 5.2 BENCHMARK EVALUATION

**Lifetime conditioning enables adaptation to variable training horizons.** In order to investigate the adaptation to variable training horizons, we evaluate LPG and TA-LPG over a range of horizons on a set of held-out Grid-Worlds (Figure 1). We observe consistently higher performance in final return from TA-LPG across all horizons. This includes short horizons—with TA-LPG reaching maximum performance on the "sparse" task in $1/8$ of the training steps required for LPG—and long horizons, such as in the "dense" task where TA-LPG continues to improve performance after LPG has plateaued.

Examining the training curves for different horizons on each task, we note that the performance of TA-LPG with longer training horizons is typically dominated by TA-LPG with shorter horizons at each step. However, these long-horizon runs then achieve a higher return beyond the short horizon, surpassing their final return. This reflects a desired mode of learning discovered by these models, being the sacrifice of intra-training performance to produce a superior final return (c.f. exploration vs. exploitation trade-off).

We evaluate TA-LPG against LPG with a heuristic learning rate schedule in Appendix C. The heuristic schedule has an insignificant impact on LPG's performance, with TA-LPG again outperforming it across training horizons. This demonstrates the insufficiency of combining static objective functions with heuristic schedules, in contrast to the success of temporally-aware objective functions.

**Lifetime conditioning improves performance on out-of-distribution environments.** Figure 2 shows that TA-LPO significantly outperforms both LPO and PPO on a wide range of tasks from the MinAtar and Brax environment suites. Notably, LPO and TA-LPO were only meta-trained on Space Invaders from the MinAtar suite. While the MinAtar PPO hyperparameters train on 10M environment steps with rollouts of length 128, the Brax hyperparameters train on 50M environment steps with rollouts of length 5, showing that the learned policy objective is robust to different hyperparameters. Interestingly, the performance of TA-LPO often only exceeds PPO towards the end of training, as seen in Appendix B, implying that PPO converged too early. Early convergence in RL is a common failure mode of RL algorithms (Nikishin et al., 2022; Lyle et al., 2022). These results potentially suggest that lifetime awareness could be one approach to addressing this issue.

## 5.3 ANALYSIS OF LEARNED UPDATE

**Adaptation to training horizon.** In Figure 3, we show the policy entropy and update norm of policies trained with TA-LPG and LPG over a range of training horizons. We observe adaptations from TA-LPG in both of these metrics as the training horizon increases. Policy entropy decays over training with both models, but TA-LPG decays entropy at a slower rate as the training horizon increases, before reaching a similar minimum entropy (beyond horizons of $2^5$ steps). This reflects balanced exploration from these models, by maintaining high entropy until late in training, regard-

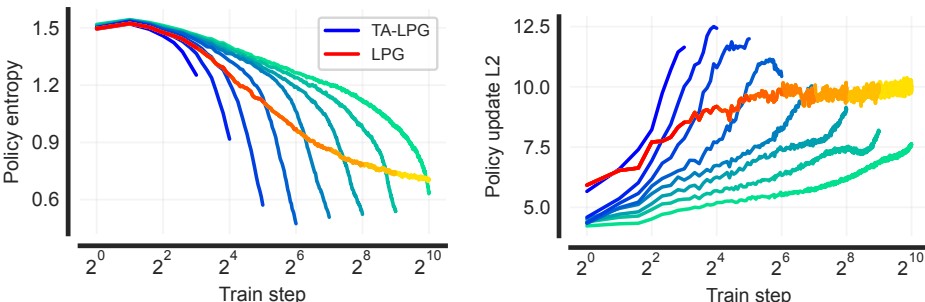

Figure 3: **Lifetime conditioning enables adaptation to training horizon.** Policy entropy and update norm of TA-LPG and LPG over randomly-sampled Grid-Worlds.

less of horizon. In contrast, LPG learns a horizon-invariant update, which induces a static entropy annealing schedule. Due to this, the maintenance of an exploratory training policy for long-horizon tasks requires the sacrifice of performance on short-horizon tasks. We observe similar behavior when analyzing the policy update norm, with both the average update norm and rate of increase in norm inversely proportional to the training horizon.

**Adaptation to relative training step.** In Figure 4 we follow the analysis used for LPO (Lu et al., 2022a) and visualize the derivative of the TA-LPO objective at different points in training. The $x$-axis is the likelihood ratio $p$, the $y$-axis is the advantage of a given transition, and the color corresponds to the gradient for a transition at that point. Positive values (red) increase the likelihood of the given transition (i.e., push the likelihood to the right) and negative values (blue) decrease them. There are two key features of the objective function:

*Asymmetric Rollback Schedule.* At $n = 0$, the lower left quadrant, which corresponds to $A < 0$ and $p < 1$ (a transition with a negative advantage and a relative decrease in action probability) has a large positive region. This suggests that the discovered algorithm, instead of clipping gradients for negative advantages, performs rollback (Wang et al., 2020) – a more aggressive form of regularization that *reverses* the gradient for extreme ratios. Meanwhile, in the top right quadrant, which corresponds to $A > 0$ and $p > 1$, the region is still positive, suggesting that the algorithm does not perform clipping at all for positive advantages. These findings align closely with the original LPO analysis (Lu et al., 2022a) and suggest an optimistic objective. However, at $n = N$, we see the *exact opposite*. There is rollback for *positive* advantage and no clipping for negative advantage. This corresponds to a high level of *risk aversion* by the algorithm at the end of training.

*Implicit Entropy Regularization Schedule.* At $n = 0$, the gradient is, on average, more positive than negative, throughout the plot. This corresponds to entropy maximization because it encourages the agent to increase the probability of all sampled actions. At $n = N$ the opposite holds, which can be viewed as *minimizing* entropy.

Overall, the analysis suggests that TA-LPO behaves similarly to LPO at $n = 0$ (it is optimistic and maximizes entropy), but over time becomes pessimistic and discourages entropy.

**Meta-gradient optimization fails to learn an adaptive update.** As outlined in Section 4.2, Oh et al. (2020) optimize LPG using meta-gradients, by backpropagating through a truncated window of updates. Since this proxy objective is myopic, we hypothesize that LPG is unable to effectively exploit temporal information when optimized with meta-gradients. To evaluate this, we trained TA-LPG with meta-gradients and evaluated against the ES-trained method over a distribution of random Grid-Worlds (Figure 5). We observe a lower return from the meta-gradient model at all training horizons, in addition to no sign of horizon adaptation, in contrast to the ES-optimized model. To further analyze the learned update, we plot the L2-norm of the agent update when trained with the meta-gradient model over a range of training horizons. Whilst the update norm responds to the relative train step $n/N$, the schedule is very similar for each evaluated horizon, with no consistent sign of adaptation. These results demonstrate the inability of meta-gradient methods to learn temporal awareness and potentially suggest that truncated meta-optimization is insufficient for discovering long-sighted algorithms.

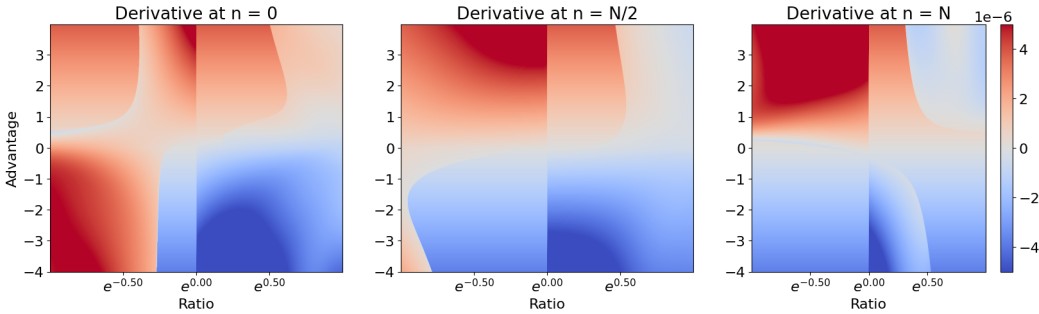

Figure 4: **TA-LPO learns to switch from optimism to pessimism.** A visualization of the derivative of the TA-LPO objective at the beginning ($n = 0$, left), middle ($n = N/2$, center), and end ($n = N$, right) of the training lifetime. The objective at $n = 0$ appears to be optimistic and maximize entropy while the objective at $n = N$ appears to be pessimistic and minimize entropy.

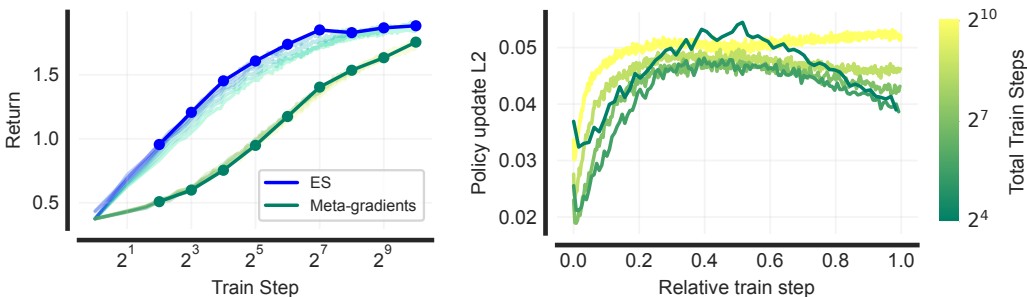

Figure 5: **Meta-gradient TA-LPG fails to adapt to temporal information. Left:** Final return of TA-LPG trained with ES and meta-gradients on the meta-training distribution of randomly sampled Grid-Worlds. Individual training curves are plotted for each horizon (faint lines), but are indistinguishable for meta-gradient TA-LPG due to lack of horizon adaptation. **Right:** Policy update norm of the meta-gradient TA-LPG model over inner loop training, for a range of training horizons. We observe lower meta-gradient performance across horizons and no consistent sign of horizon adaptation in update norm.

## 6    CONCLUSION

**Summary**    We introduced a novel family of parameterized RL objective functions that are *temporally aware*. After meta-training, the learned objective functions demonstrate adaptation over an agent's lifetime, resulting in strong generalization to unseen training horizons and environments. Furthermore, we showed that they incorporate adaptive motifs that change throughout learning, with the high-level behavior demonstrating a shift from optimism to pessimism over the training process. Finally, we demonstrated evolutionary optimization as a critical factor for discovering RL algorithms capable of effective lifetime conditioning.

**Limitations**    We show that the discovered temporally-aware algorithms are capable of broad generalization to both discrete pixel-based environments and continuous control settings as well as various time horizons. Nonetheless, for LPG it is difficult to obtain strong (theoretical) robustness guarantees. Such robustness would benefit reliable deployment. On the other hand, LPO meta-learns in a much more restricted space than LPG and may be less expressive.

**Future Work**    Our contributions highlight the importance of expressivity in discovered RL algorithms. Here, we only considered conditioning on temporal information. In the future, we believe that further performance gains can be achieved by augmenting algorithms with more information such as optimizer statistics, network architectures, and task information. Eventually, we may be able to remove these handpicked augmentations and perform RL purely in-context with blackbox meta-learning (Lu et al., 2023a; Kirsch et al., 2023).

ACKNOWLEDGMENTS

Matthew Jackson is funded by the EPSRC Centre for Doctoral Training in Autonomous Intelligent Machines and Systems, and Amazon Web Services.

ETHICS STATEMENT

We propose a method for the automated discovery of time-aware RL algorithms that employ gradient-based learning on flexible meta-learned loss functions. Like in other meta-learning approaches, learning-to-learn automates and thus reduces our knowledge about the learning process. This requires additional monitoring and introspection, such as the analysis we have provided in Figure 4. The process is aided by evolutionary methods and can potentially result in a system that improves its own performance throughout the course of learning.

REPRODUCIBILITY STATEMENT

The method presented in our paper can be performed with a small modification to an implementation of LPO and LPG. We describe this modification in Section 4 and our hyperparameters in Appendix A. Our implementation of LPG, LPO, and their temporally-aware modifications (TA-LPG, TA-LPO) can be found at https://github.com/EmptyJackson/groove.

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

# A HYPERPARAMETERS

## A.1 LPG

Table 1: LPG hyperparameters

| Hyperparameter | Value |
|---:|---|
| Optimizer | Adam |
| Learning rate | 1e-4 |
| Discount factor | 0.99 |
| Number of interactions per agent update | 20 |
| Number of parallel lifetimes | 512 |
| Number of parallel environments per lifetime | 64 |
| Policy entropy coefficient ($\beta_0$) | 0.05 |
| Bootstrap entropy coefficient ($\beta_1$) | 0.001 |
| L2 regularization coefficient for $\hat{\pi}$ ($\beta_2$) | 0.005 |
| L2 regularization coefficient for $\hat{y}$ ($\beta_3$) | 0.001 |
| Number of agent updates per optimizer update | 5 |
| ES Learning Rate Decay | 0.999 |
| ES Learning Rate Limit | 1e-5 |
| ES Sigma Init | 0.003 |
| ES Sigma Decay | 1.0 |
| ES Sigma Limit | 0.001 |

## A.2 LPO

The drift function is parameterized by a one-layer fully-connected network with 1 hidden layer and 128 hidden units. Meta-training was done on 2 A100 GPUs with synchronous updates.

Table 2: Important parameters for Training LPO and PPO

| Hyperparameter | Value |
|---|---|
| **Meta-Evolution** | |
| Population Size | 64 |
| Number of Hidden Layers | 1 |
| Size of Hidden Layer | 128 |
| Number of Generations | 128 |
| Centered Ranking | True |
| ES Sigma Init | 0.04 |
| ES Sigma Decay | 0.999 |
| ES Sigma Limit | 0.01 |
| **MinAtar** | |
| Number of Timesteps | 1e7 |
| Number of Environments | 64 |
| Unroll Length | 128 |
| Number of Minibatches | 8 |
| Number of Update Epochs | 4 |
| Learning Rate | 5e-3 |
| Gamma | 0.99 |
| Max Grad Norm | 8.0 for LPO, 0.5 for PPO |
| Entropy Coefficient | 0.01 |
| **Brax** | |
| Number of Timesteps | 5e7 |
| Number of Environments | 2048 |
| Unroll Length | 10 |
| Number of Minibatches | 32 |
| Number of Update Epochs | 4 |
| Learning Rate | 3e-4 |
| Gamma | 0.99 |
| Max Grad Norm | 8.0 for LPO, 0.5 for PPO |
| Entropy Coefficient | 0.0 |

# B  TA-LPO RESULTS

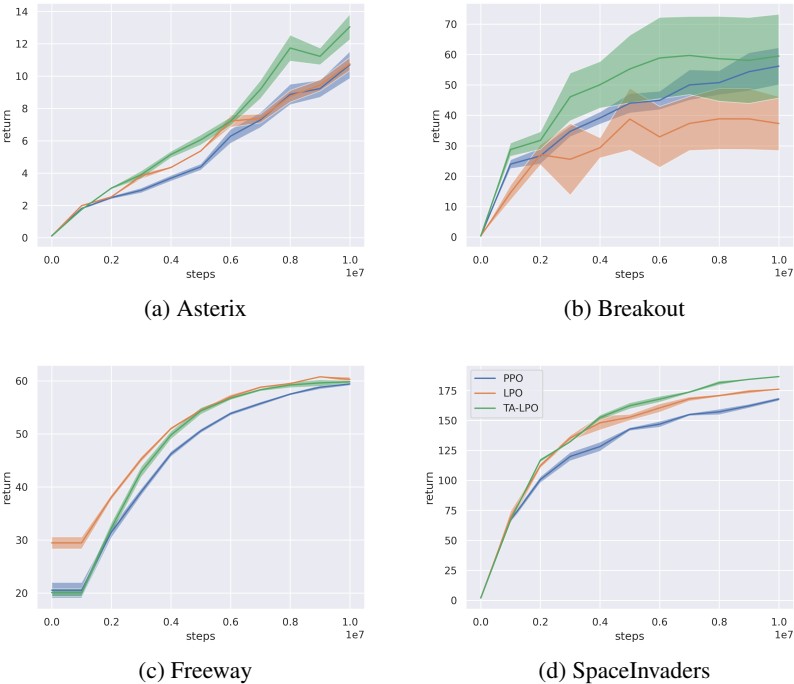

(a) Asterix

(b) Breakout

(c) Freeway

(d) SpaceInvaders

Figure 6: Performance of PPO, LPO, and TA-LPO on the Minatar environments. The curves show the mean evaluation return across 3 random seeds, with error bars showing standard error of the mean. LPO and TA-LPO were meta-trained on SpaceInvaders.

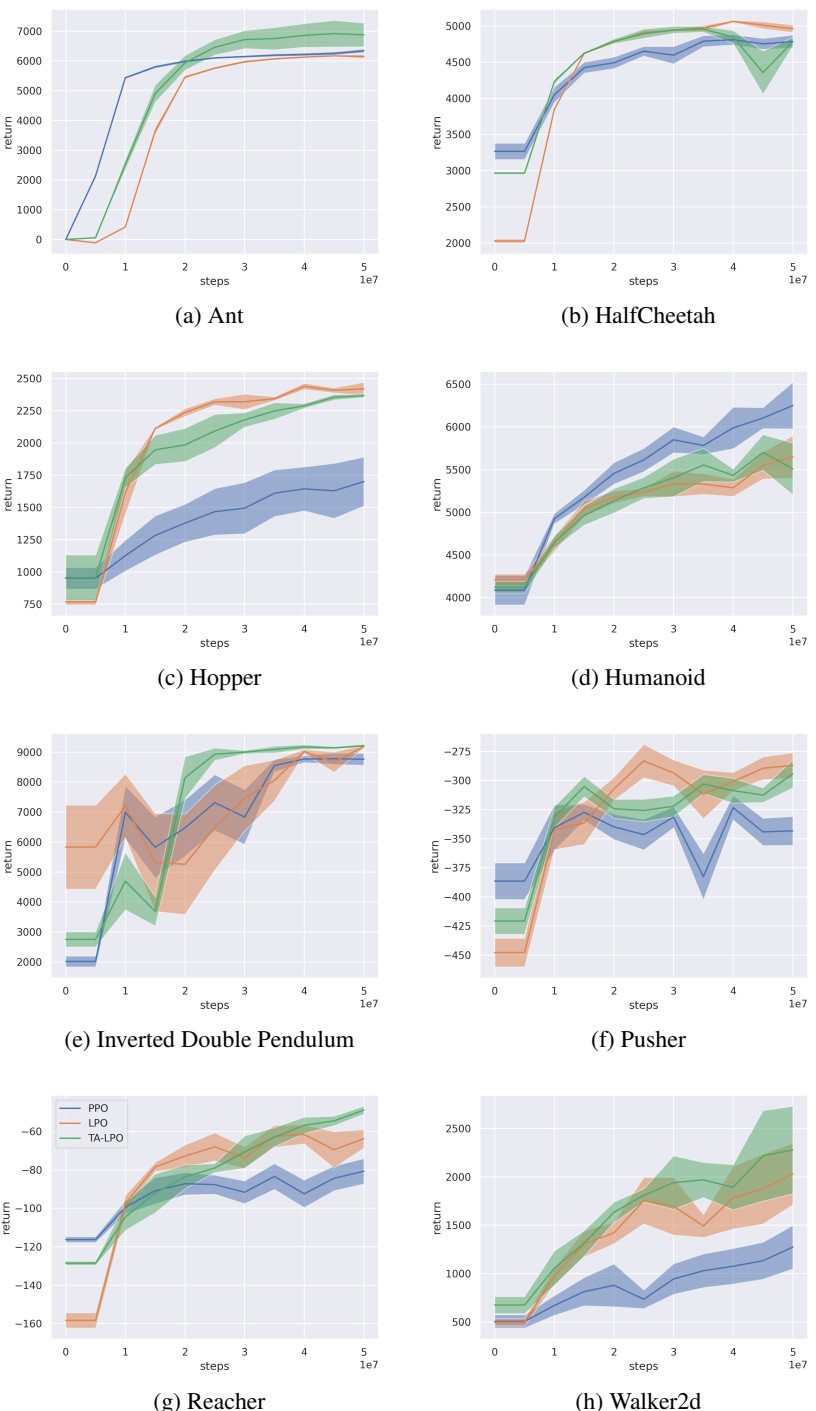

Figure 7: Performance comparison between PPO (blue), LPO (orange), and TA-LPO (green) in Brax environments. The curves show the mean evaluation return across 3 random seeds, with error bars showing standard error of the mean. LPO and TA-LPO were not meta-trained on any of these environments.

## C  HEURISTIC ANNEALING SCHEDULES

As a heuristic baseline for horizon adaptation, we augment LPG with a learning rate scheduler and compare performance against TA-LPG (Figure 8). As with the original LPG, TA-LPG consistently outperforms the scheduler at all training horizons. Furthermore, we observe that the the performance of LPG with learning rate scheduling is insignificantly different from its original performance (Figure 1). This highlights the failure of the insufficiency of heuristic temporal adaptation methods and the significance of the method learned by TA-LPG.

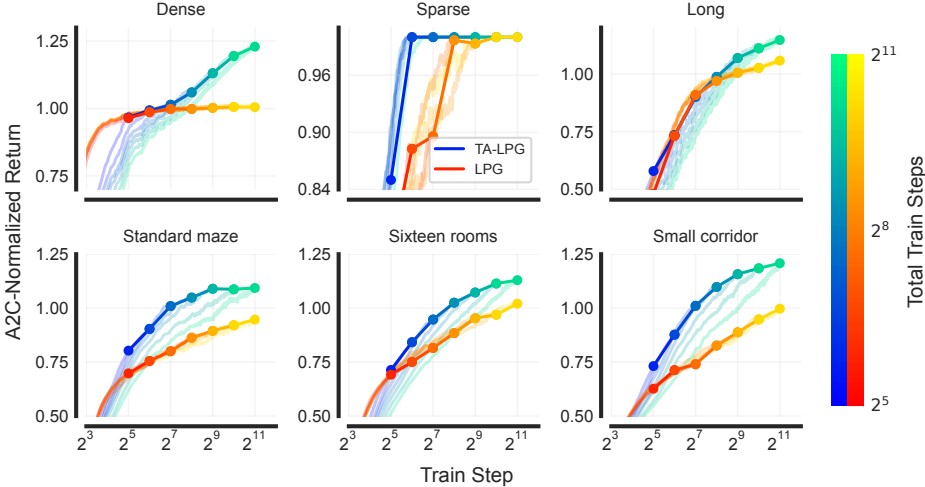

Figure 8: Training curves for LPG with a tuned learning rate scheduler and TA-LPG.

# D HYPERPARAMETER SEARCH OVER ENTROPY

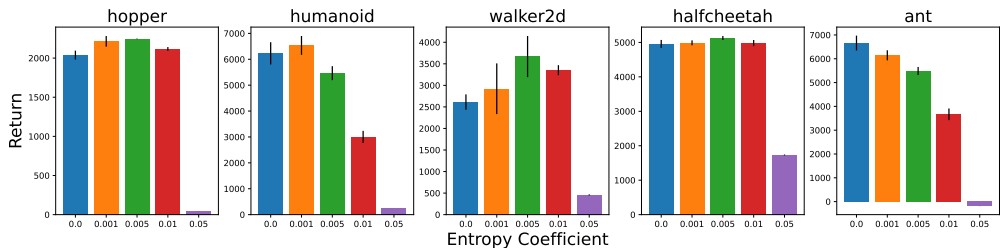

Figure 9: Hyperparameter Sweep for 5 Brax environments over the entropy coefficient.

