# D   HYPERPARAMETER SEARCH OVER ENTROPY

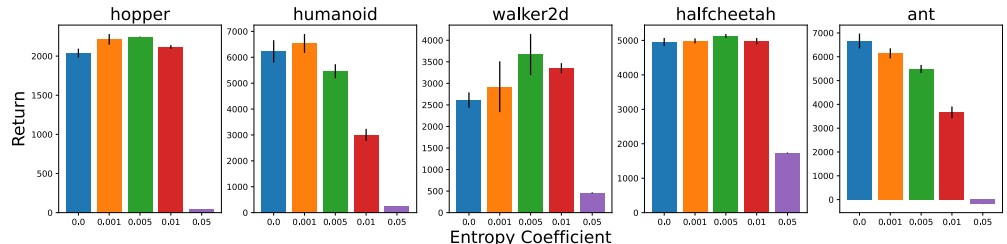

Figure 9: A hyperparameter sweep over the entropy coefficient. The error bars represent the standard error across five seeds at the end of training. We observe that a coefficient of 0.0 performs comparably to the other ones, and no single entropy coefficient is strictly better than 0.0.

# E  REQUIREMENT FOR TEMPORAL AWARENESS IN REINFORCEMENT LEARNING

In this work, we propose conditioning learned policy optimizers on temporal information about the agent's learning progress. Here, we present theoretical demonstrations for why temporal conditioning is required for optimality, given insufficient training coverage or a stochastic environment.

For our proof sketches, we consider a distribution of two-arm bandits $\mathcal{B}$ with state space $\mathcal{S} = \{s\}$ and action space $\mathcal{A} = \{a_1, a_2\}$. We define the expected reward for the two arms as random variables $R_1$ and $R_2$, whose expectations $\mathbb{E}_{b \sim \mathcal{B}}[R_1]$ and $\mathbb{E}_{b \sim \mathcal{B}}[R_2]$ across $\mathcal{B}$ are known by the optimizer. After $n$ training interactions, the policy optimizer $\mathcal{F}$ returns a policy $\pi_n = \mathcal{F}(\tau_{:n})$ conditioned on all prior environment interactions $\tau_{:n}$. The optimizer performs $N$ training interactions, after which $\pi_N$ is returned and evaluated as the final policy. Informally, the policy optimizer is optimal if the expected return of $\pi_N$ over $\mathcal{B}$ is maximal, given $N$ environment interactions.

**Theorem 1 (Deterministic MDPs with insufficient coverage)** *Given a deterministic MDP and an insufficient number of training interactions $N$ to sample all state-action pairs, temporal awareness (knowledge of $N$) is required for the policy optimizer to be optimal.*

**Proof Sketch**  Consider a distribution of deterministic two-arm bandits $\mathcal{B}_{\text{det}}$, such that the rewards for each arm $r_1$ and $r_2$ are deterministic. Proceeding by counterexample, we assume our policy optimizer $\mathcal{F}$ is optimal over $\mathcal{B}_{\text{det}}$, for all values and without knowledge of the total number of training interactions $N$. Without loss of generality, let the first interaction select action $a_1$ and observe reward $r_1$. Proceeding by cases, if $N = 1$ and $r_1 > \mathbb{E}[r_2]$, then the policy returned by the optimizer $\pi_1$ should deterministically select $a_1$, since it will achieve higher reward in expectation. However, if $N = 2$, then $\pi_1$ should deterministically select $a_2$, in order to sample $r_2$ and infer the optimal policy after this interaction. This is a contradiction, so $\mathcal{F}$ is not optimal for all values and without knowledge of $N$.

**Theorem 2 (Stochastic MDPs)** *Given a stochastic MDP, temporal awareness (knowledge of $N$) is required for the policy optimizer to be optimal.*

**Proof Sketch**  Consider a distribution of stochastic two-arm bandits $\mathcal{B}_{\text{rand}}$, such that the rewards sampled from each arm $r_1$ and $r_2$ are stochastic. Proceeding by counterexample, we assume our policy optimizer $\mathcal{F}$ is optimal over $\mathcal{B}_{\text{rand}}$, for all values and without knowledge of the total number of training interactions $N$. After $n$ interactions with $\mathcal{B}_{\text{rand}}$, we assume our optimizer can compute the posterior distribution over the expected rewards, given by $p(R_1|\tau_{:n})$ and $p(R_2|\tau_{:n})$. Without loss of generality, let $\mathbb{E}[R_1|\tau_{:n}] > \mathbb{E}[R_2|\tau_{:n}]$, such that taking action $a_1$ has a higher expected reward than $a_2$ given the observed interactions $\tau_{:n}$. Proceeding by cases, if $N = n$, then the policy returned by the optimizer $\pi_n$ should deterministically select $a_1$, since its expected reward is higher. However, if $N > n$, then $\pi_n$ should remain stochastic in general, such that the belief over the expected reward for each arm can be refined. This is a contradiction, so $\mathcal{F}$ is not optimal for all values and without knowledge of $N$.

## F ALTERNATIVE TEMPORAL REPRESENTATIONS

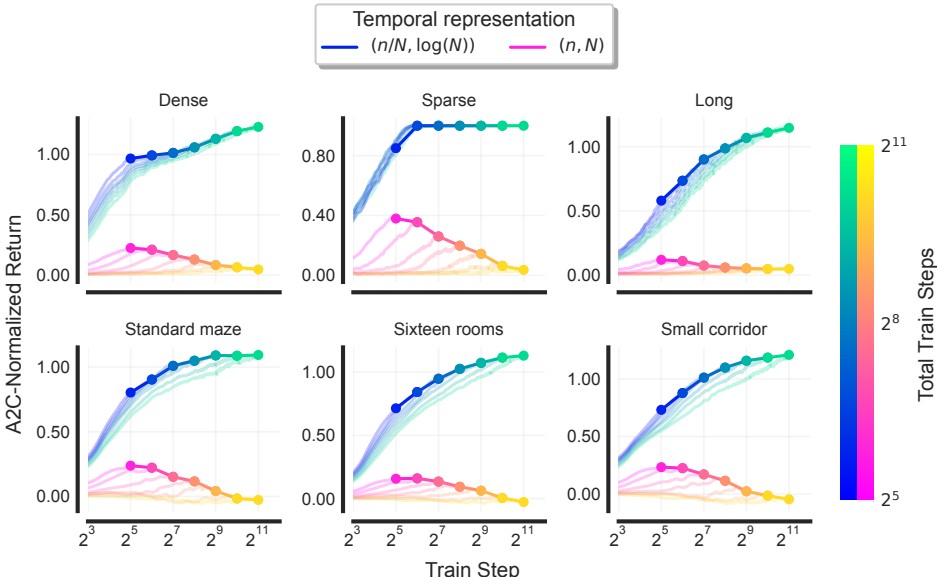

Figure 10: **TA-LPG requires a suitably transformed temporal representation.** Training curves for TA-LPG on held-out Grid-World environments, with the original temporal representation $(n/N, \log(N))$ and direct conditioning on the current and total timesteps $(n, N)$. We observe poorer performance from directly conditioning on temporal information without transformation, particularly for tasks with high training budgets $N$. This is as expected, since direct conditioning leads to unbounded and linearly scaling values, unlike our transformed representation.

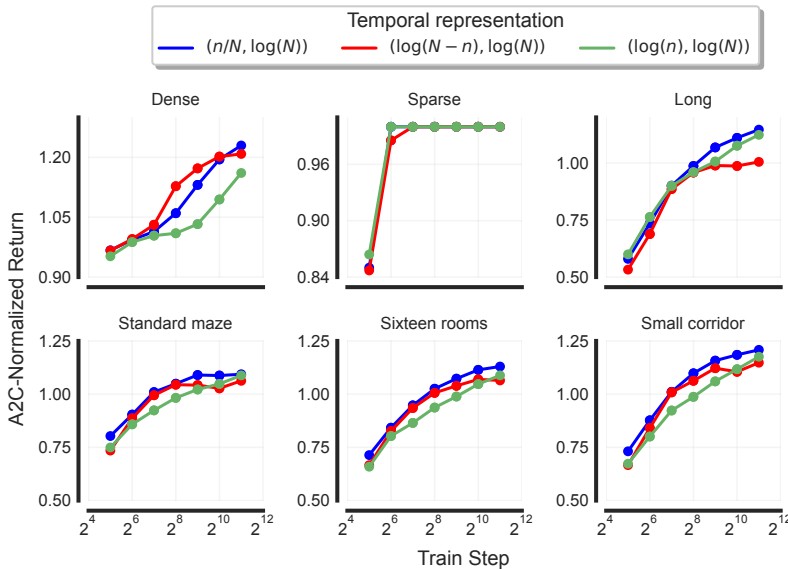

Figure 11: **TA-LPG is robust to alternative transformed temporal representations.** Final performance for TA-LPG on held-out Grid-World environments over a range of training horizons, with the original temporal representation $(n/N, \log(N))$ and two alternatively transformed representations, $(\log(N - n), \log(N))$ and $(\log(n), \log(N))$. Individual training curves are omitted for clarity. We observe no significant difference in performance between representations. This demonstrates the need for logarithmic scaling or bounding in the temporal representation, as well as TA-LPG's robustness under these constraints.