# OpenReview forum: "Discovering Temporally-Aware Reinforcement Learning Algorithms"
_ICLR.cc/2024/Conference — ICLR 2024 poster_

### Official Review · Reviewer_hAap · 2023-10-18

**Soundness:** 4 excellent
**Presentation:** 3 good
**Contribution:** 3 good
**Rating:** 8
**Confidence:** 4

**Summary:**

The paper investigates the problem of discovering reinforcement learning algorithms via meta-learning. The paper reveals that incorporating temporal data regarding the agent's lifetime empowers the newly discovered algorithm to dynamically refine its objectives, such as the exploration-exploitation tradeoff, thereby fostering the development of a higher-performing agent. The proposed method can be readily combined with two existing algorithm, LPG and LPO, and demonstrates its effectiveness on a wide variety of benchmarks, including MiniGrid and MinAtar.

**Strengths:**

- Novelty: Although the proposed method is a simple modification of previous algorithms, the idea of utilizing temporal information is well-motivated and novel.
- Presentation: The authors meticulously analyze the evolution of the discovered objectives throughout the agent's lifetime, detailed in Section 5.3, with Figures 3 and 4 providing compelling visual representations of these dynamic shifts.
- Experiment: The authors meticulously design experiments to rigorously assess the generalization ability of the proposed method to different environments or training hyperparameters (e.g. the number of environment interactions).

**Weaknesses:**

- Novelty: The application of antithetic sampling for gradient estimation, while prevalent in black-box optimization, appears to lack originality.
- Ablation: The authors do not conduct a comprehensive ablation study comparing the proposed TA-LPG with the standard LPG. Specifically, they introduce two significant modifications, incorporating temporal information and employing antithetic sampling, without isolating the effects of each change.

**Questions:**

- Ablation: Could you please provide me a clear understanding of the individual contributions of the two proposed component to the performance improvements?
- Hyperparameter: Setting the entropy coefficient of PPO to 0.0 in the Brax environment appears to significantly limit exploration in my view and poses questions about the experimental design. To address this and strengthen the validity of your results, I recommend conducting a systematic hyperparameter search on the entropy coefficient of PPO within the Brax environment.

---

> ### Author Response · Authors · 2023-11-17
>
> We would like to thank the reviewer for their thoughtful and insightful review. We are glad the reviewer finds our work “well-motivated and novel”, with “compelling visual representations” and “meticulous” experiments. The reviewer mentions several good points that we would like to address.
>
> ## Novelty of antithetic task sampling
>
> You are correct, using antithetic sampling for ES is not a novelty (Salimans et al., 2017). However, the update we propose in (12), “antithetic task sampling”, is designed to reduce the variance from applying ES in the multi-task setting. Using the update from Salimans et al. (2017), each candidate is evaluated on a randomly sampled task, before a rank transformation is applied to their fitness. This can lead to instability in the update, when the fitness across tasks has varying scales. In (12), we propose evaluating each antithetic candidate pair *on the same task*, before applying a rank transformation over the pair (equivalent to selecting the higher-performing candidate). This allowed us to normalize fitness across tasks, which we found led to faster meta-training convergence and improved final performance.
>
> ## Ablation of antithetic task sampling
>
> We do not ablate antithetic task sampling in the paper since it is tangential from the focus of the work. However, we include it since it is a novel and impactful implementation detail. We note that ES with antithetic task sampling is applied to both LPG and TA-LPG in all experiments (other than the meta-gradient model), so the ablation of temporal information is independent of this detail and rigorously demonstrates the impact of temporal information.
>
> ## Hyperparameters
> This is a good discussion point. Our reasoning for setting entropy=$0.0$ is based on [OpenAI's original PPO implementation](https://github.com/openai/baselines/blob/master/baselines/ppo2/defaults.py), which uses that setting for continuous control. [Extremely thorough follow-up work](https://arxiv.org/abs/2006.05990) (See Decision C32, Figure 77) [1] has found no evidence that entropy significantly improves performance on continuous control tasks. While we can tune this coefficient per-environment, we believe that this is against the spirit of the benchmark and is not commonly done.
>
> For a more in-depth discussion of this, we recommend [this blog post](https://iclr-blog-track.github.io/2022/03/25/ppo-implementation-details/) [2].
>
> Nonetheless, we have run the requested set of experiments and have done a hyperparameter sweep over the entropy coefficient in similar settings to [1] with 5 seeds over 5 values and have attached the results to the supplementary material in Figure 9. In general, our findings align with [1] and we find that an entropy coefficient of 0.0 performs comparably to the others. While an entropy coefficient of 0.005 performs significantly better in Walker2d, it performs significantly worse than 0.0 in Ant and Humanoid. If the reviewer believes this would significantly improve the analysis of our paper, we can provide updated figures that tune the coefficient per-environment and per-algorithm.
>
> [1] Andrychowicz, Marcin, et al. "What matters for on-policy deep actor-critic methods? a large-scale study." International conference on learning representations. 2020.
>
> [2] Huang, et al., "The 37 Implementation Details of Proximal Policy Optimization", ICLR Blog Track, 2022.
>
> ---
>
> *We hope that most of the reviewer’s concerns have been addressed and, if so, they would consider updating their score. We’d be happy to engage in further discussions.*

---

> ### Comment · Reviewer_hAap · 2023-11-21
> **Reply**
>
> Thank you for the comprehensive response. I was concerned that antithetic sampling might only be applied to TA-LPG. I encourage the authors to make this more explicit in the paper. Since all of my concerns have been well addressed, I raise the score from 6 to 8.

---

### Official Review · Reviewer_c7mC · 2023-10-26

**Soundness:** 3 good
**Presentation:** 3 good
**Contribution:** 3 good
**Rating:** 8
**Confidence:** 4

**Summary:**

This work proposes to meta-learn a better objective function for reinforcement learning (RL) tasks by taking into account the information of the task horizon.
Specifically, with the help of the time-step information, the proposed algorithms could find a better balance between exploration and exploitation.
Furthermore, this work shows that meta-gradient methods fail to adapt to different horizons, while evolution strategies can do better in this case.

**Strengths:**

The idea behind this work is simple and effective, supported by solid experiments and detailed analysis.

**Weaknesses:**

- The idea of incorporating time-step information is not novel in RL, such as this [work](https://proceedings.mlr.press/v80/pardo18a.html), which should be included in the related work.

- Lack of ablation study: In Section 4.1 Equation 9, $n/N$ and $\log(N)$ are included as part of the input to the agent. It is not clear or argued how good this option is compared with other options, such as $n$ and $N$, $N-n$, or $n/N$.

**Questions:**

- Section 4.1, after Equation 10, "Since TA-LPO is only meta-trained on a single environment and horizon, we do not include the $\log(N)$ term since it does not vary across updates." Would including the $\log(N)$ term decrease the performance?
- Section 5.2, Paragraph "Lifetime conditioning improves performance on out-of-distribution environments." Figure Figure 2: typo.

---

> ### Author Response · Authors · 2023-11-17
>
> We would like to thank the reviewer for their clear and useful review. We are glad the reviewer finds our work to be “simple and effective” and “supported by solid experiments and detailed analysis”. The reviewer brings up many good suggestions and questions we would like to address.
>
> ## Time Limits in RL
>
> We would like to thank the reviewer for pointing out this line of related work. We have updated the manuscript to include the suggested paper. However,  there is an important difference in the use of the term “time-step” between these works. In our work, “time-step” refers to the total training budget (i.e. a “global” time-step). For the mentioned work, “time-step” refers to the time-step *within* an episode. We’ve further updated the manuscript to clarify this.
>
> Even with this distinction, we agree that incorporating time step information is not novel in RL. Any learning rate schedule incorporates global time-step information. However, incorporating time-step information in RL algorithm discovery methods  (e.g. LPG and LPO) is novel, and leads to the significant improvements in performance demonstrated in our work.
>
> ## Ablation on the input
>
> Thank you for raising this question, we have included a discussion of alternative representations in the revised paper and are running the ablations outlined below.
>
> For LPG, our choice of representation was fairly arbitrary. We decided to include $\log(N)$ as a measure of total training budget - applying a logarithmic transformation in order to handle large values - and $n/N$ as a measure of training progress that would be invariant to $N$. We did not condition on $N$ or $n$ directly, since these numbers can be very large (e.g. $1e7$ for Min-Atar) and lead to dramatically out-of-distribution values when transferring to much longer or shorter horizons. As long as we have a measure of both total and relative train step, and apply suitable transformations, we are confident that the representation of total and relative train step will have a negligible impact on performance.
>
> To verify this, we are rerunning the experiments and will provide results with alternative parameterisations for training progress - $\log(N-n)$ and $(\log(N), \log(n))$ - in addition to another experiment directly conditioning on the total train steps, $N$.
>
> > Would adding $\log(N)$ to TA-LPO decrease performance?
>
> Since TA-LPO is trained on a single task with a constant $N$, $\log(N)$ would simply be a constant bias term. Therefore, adding $\log(N)$ to TA-LPO would be unlikely to impact in-distribution performance. However, including $\log(N)$ could affect transfer to out-of-distribution tasks with different $N$, since the input would then be out of distribution.
>
> > Figure 2 typo
>
> Thank you! We’ve fixed this in our updated manuscript.
>
> ---
> *We hope that most of the reviewer’s concerns have been addressed. We’d be happy to engage in further discussions.*

---

> > ### Comment · Reviewer_c7mC · 2023-11-20
> > **Reply**
> >
> > Thank you for your reply. Please let me know when you have the results of the ablations.

---

> > > ### Author Response · Authors · 2023-11-21
> > > **New results**
> > >
> > > We have updated our revision to include new experiments with alternative temporal representations (Appendix F), which can also be found with other new results in the supplementary material. As expected, these results verify our hypothesis that TA-LPG is robust to the representation of temporal information, under the condition that it is suitably scaled or bounded.
> > >
> > > We hope this addresses your concern regarding ablations and thank you again for the suggestion!

---

### Official Review · Reviewer_47vA · 2023-10-29

**Soundness:** 3 good
**Presentation:** 2 fair
**Contribution:** 3 good
**Rating:** 5
**Confidence:** 5

**Summary:**

This paper introduces a training horizon to meta-reinforcement learning algorithms to discover objective functions that depend on the learner's lifetime. The basic idea is to add the information about the lifetime to the input vector of the meta-learning algorithm. The authors propose two algorithms, Temporally-Adaptive Learned Policy Gradient (TA-LPG) and TA-Learned Policy Optimization (TA-LPO) by extending LPG and LPO, respectively. Then, the authors found that the evolutionary strategy is more appropriate than meta-gradient approaches to optimize the upper-level objective function.

**Strengths:**

Based on the proposed idea, the authors implement two meta-reinforcement learning algorithms: TA-LPG and TA-LPO. It implies that the idea can be applied to various algorithms, although input augmentation depends on the algorithms. Systematic experiments show that TA-LPG and TA-LPO outperform LPG and LPO, respectively.

**Weaknesses:**

Although the experimental results support the authors' hypothesis that conditioning on the lifetime is helpful in meta-learning, there are no theoretical justifications. In addition, the way to incorporate the lifetime depends on the algorithm, and the experimenter carefully has to design augmentation.

**Questions:**

1. The function $U_\phi (x_t \mid x_{t+1}, \ldots, x_T)$ is defined in Section 3.2, but it is unclear because $T$ is not explained. Is it the total number of interactions, denoted by $N$ later?
2. Two variables $n/N$ and $\log N$ are augmented in TA-LPG. Is $n/N = 0$ when $N$ is unbounded? In my understanding, $N$ is determined and fixed before training. Would you explain $N$ and provide an example task?
3. In TA-LPO, $ \frac{n}{N} x_{r, A}$ is augmented. It is not straightforward to me because it is proportional to $x_{r, A}$, which means linearly-dependent. Would you explain the problem if $n/N$ and $\log N$ are added as the authors did in TA-LPG?
4. In the final paragraph of Section 4.2, the authors mentioned, "We found this stabilized training and led to higher performance..." Would you explain "this stabilized training" in detail? If it means the rank transformation, it is used in Salimans et al. (2017). Discussing the relation between two equations would be better if the stabilized training means Eq. (12) rather than (11).
5. I do not fully understand the final paragraph of Section 3.1. In the manuscript, LPG and LPO are selected as the base algorithms. However, the authors mentioned as follows: In our work, we focus on instances of meta-RL that parameterize surrogate loss functions with $\phi$ and apply gradient-based updates to $\pi_\theta$ (Houthooft et al., 2018; Kirsch et al., 2019; Bechtle et al., 2021). Does it mean that three algorithms are implemented somewhere?
6. Equation (10): Is $x_{r, A}$ a typo of $x_{p, A}$?
Please update the reference Kirsch et al. (2019) to Kirsch et al. (2020). Please see https://openreview.net/forum?id=S1evHerYPr

---

> ### Author Response · Authors · 2023-11-17
>
> We would like to thank the reviewer for their very thorough and detailed review. We are glad that the reviewer finds that our experiments are systematic and show the strength of our method.
> The reviewer brings up several good points and questions that we would like to address.
>
> ## On theoretical justifications
>
> We’ve included a theoretical justification and proof in Appendix E. The high-level justification is that without temporal awareness, the algorithm cannot effectively trade off exploration and exploitation. For example, in the case of multi-armed bandits, the agent should converge to a single lever at the end of training, but perform exploration during training.
>
>
> ##  On “careful” augmentations
>
> For LPG, our choice of augmentation was arbitrary. We decided to include $\log(N)$ as a measure of total training budget - applying a logarithmic transformation in order to handle large values - and $n/N$ as a measure of training progress that would be invariant to $N$. As long as we have a measure of each of these quantities and are scaled to a reasonable range, we expect the representation of total and relative train step to have a negligible impact on performance.
>
> To verify this, we are rerunning the experiments and will provide results with alternative parameterisations for training progress - $\log(N-n)$ and $(\log(n))$ - in addition to another experiment directly conditioning on the total train steps, $N$.
>
> > Would you explain the problem if $n/N$  and $\log(N)$ are added [to TA-LPO]?
>
> For TA-LPO it is *slightly* more involved. Given a quick understanding of the theoretical framework of LPO, TA-LPO is a simple augmentation.
>
> The theoretical framework behind LPO is described in Section 3.3. The key takeaway from is that the network needs to output $0$ when the likelihood ratio $p$ is equal to $1$. The network used for LPO has no biases and uses a tanh activation, so if the input is $0$, then the output is $0$. Thus, the input to the LPO network, $x_{r,A}$, is designed to be $0$ when $p=1$ (Equation 7).
>
> If we simply append $n/N$ as an input to TA-LPO, then the requirement doesn’t hold since it is not $0$ when $p=1$. Thus, we multiply $n/N$ by $x_{r,A}$ (which is $0$ when $p=1$).
>
> We do not include any variant of $\log(N)$ because LPO is only trained with a single $N$, which would simply reduce that term to a bias term.
>
> We believe this is an insignificant challenge for future work and does not reduce the strength of our contribution.
>
> ## Questions:
>
> > $T$ is not explained
>
> Thank you for catching this. $T$ is the trajectory/rollout length that the actor collects each update step, not the total number of environment interactions $N$. We have updated the manuscript to include this!
>
> > Is $n/N$ unbounded when $N$ is unbounded? Would you explain $N$ and provide an example task?
>
> $N$ is the total number of environment interactions we provide to the agent to learn. For example, in MinAtar-Breakout, the agent interacts with the environment for $1e7$ timesteps, so $N=1e7$. From this, we calculate $n/N$, which measures the proportion through training, and $\log(N)$, which provides the agent with the total number of training steps.
>
> When $N$ is unbounded, $n/N$ is a constant $0$. In practice, the total training budget $N$ is usually known, as this allows RL algorithms to trade off exploration over training.
>
> > Would you explain "this stabilized training" in detail?
>
> Thank you for pointing this out, we have revised the paper to make this clearer. The update we propose in (12), “antithetic task sampling”, is designed to reduce the variance from applying ES in the multi-task setting. Using the update from Salimans et al. (2017), each candidate is evaluated on a randomly sampled task, before a rank transformation is applied to their fitness. This can lead to instability in the update, when the fitness across tasks has varying scales. In (12), we propose evaluating each antithetic candidate pair *on the same task*, before applying a rank transformation over the pair (equivalent to selecting the higher-performing candidate).
>
> This allowed us to normalize fitness across tasks, which led to faster meta-training convergence and improved final performance. We do not examine this rigorously in the paper since it is tangential from the focus of the work. However, we include it since it is a novel and impactful implementation detail, which we had to apply consistently throughout the LPG experiments.
>
> > Does it mean that three algorithms are implemented somewhere?
>
> Thank you for pointing this out! We have updated the manuscript to clarify this. We implemented LPG and LPO, which are successors to those papers.
>
> ## Misc:
>
> Thank you for catching the typo in equation 10 and the incorrect year on the reference! We’ve fixed these in the updated manuscript.
>
> ---
> *We hope that most of the reviewer’s concerns have been addressed and, if so, they would consider updating their score. We’d be happy to engage in further discussions.*

---

> > ### Author Response · Authors · 2023-11-21
> > **New results**
> >
> > We have updated our revision to include new experiments with alternative temporal representations (Appendix F), which can also be found with other new results in the supplementary material. As expected, these results verify our hypothesis that TA-LPG is robust to the representation of temporal information, under the condition that it is suitably scaled or bounded.
> >
> > We hope these results address the reviewer's concern about the careful design of the temporal representation, and our new theoretical justification (Appendix E) addresses their concern over the lack of theory.

---

### Author Response · Authors · 2023-11-17
**Summary of rebuttal**

We are grateful to the reviewers for their insightful feedback. We appreciate the general consensus that our work clearly demonstrates that temporal-awareness significantly improves the performance of meta-learned loss functions in RL.

R1 (47vA): “Systematic experiments show that TA-LPG and TA-LPO outperform LPG and LPO, respectively.”

R2 (c7mC): “this work is simple and effective, supported by solid experiments and detailed analysis”

R3 (hAap): “​​The authors meticulously design experiments to rigorously assess the generalization ability of the proposed method”

Multiple reviewers (R1, R2) suggested that the representation of temporal information be ablated. We’ve provided an explanation of how we selected the given representation and why we believe this has little impact, and are running new experiments to demonstrate TA-LPG’s robustness to reasonable choices of representation (which we will share by Monday). We further include an explanation for why our augmentation for TA-LPO is an extremely simple and theoretically-motivated extension of LPO (see our response to R1).

R1 further mentioned that we did not provide theoretical justifications for temporal awareness. We have included a new proof in the revision (Appendix E) demonstrating the requirement for temporal awareness.

R3 had questions about our ablations and choice of hyperparameters. We provide the requested hyperparameter sweep in Figure 9 and discuss it in-depth in our response.

Finally, we’ve made numerous miscellaneous other changes to our manuscript using the helpful feedback from the reviewers, such as:
- Clarifying the definitions of $T$ and $N$ (R1)
- Clarifying the LPO augmentation (R1, R2)
- Clarifying the novelty of antithetic task sampling, in contrast to Salimans et al. (2017) (R1, R3)
- Adding relevant references (R2)
- Fixing typos (R1, R2)

---

### Meta-Review · Program_Chairs · 2024-01-15

**Metareview:**

The paper proposes to include information regarding  the training horizon to meta-reinforcement learning algorithms in order to discover objective functions that depend on the learner's remaining lifetime. The basic idea is to add the information about the lifetime to the input vector of the meta-learning algorithm. The authors found that optimization with evolutionary strategies is more effective than meta-gradient approaches for meta-learning.

**Justification For Why Not Higher Score:**

The proposed method is direct extension of existing work.  The experimental environments are overall simple.

Only comparisons to meta-RL algorithms are included, no comparisons against vanilla RL baselines.

**Justification For Why Not Lower Score:**

The reviewers find the idea of the paper novel with solid experimentation.

---

### Decision · Program_Chairs · 2024-01-16

Accept (poster)